# Long-Term SARS-CoV-2-Specific Humoral and T Cell Responses after the BNT162b2 or BBIBP-CorV Booster and the Incidence of Breakthrough Infections among Healthcare Workers

**DOI:** 10.3390/vaccines12010003

**Published:** 2023-12-19

**Authors:** Zsolt Matula, Gabriella Bekő, Viktória Király, Márton Gönczi, András Zóka, András Baráth, Ferenc Uher, István Vályi-Nagy

**Affiliations:** 1Laboratory for Experimental Cell Therapy, Central Hospital of Southern Pest, National Institute of Hematology and Infectious Diseases, 1097 Budapest, Hungary; uher.ferenc@dpckorhaz.hu; 2Central Laboratory of Central Hospital of Southern Pest, National Institute of Hematology and Infectious Diseases, 1097 Budapest, Hungary; beko.gabriella@dpckorhaz.hu (G.B.); kiraly.viktoria@dpckorhaz.hu (V.K.); gonczi.marton@dpckorhaz.hu (M.G.); zoka.andras@dpckorhaz.hu (A.Z.); barath.andras@dpckorhaz.hu (A.B.); 3Department of Hematology and Stem Cell Transplantation, Central Hospital of Southern Pest, National Institute of Hematology and Infectious Diseases, 1097 Budapest, Hungary; valyi-nagy.istvan@dpckorhaz.hu

**Keywords:** SARS-CoV-2, long-term immunity, anti-S1/S2 IgG, anti-S IgA, anti-RBD IgG, T cell response, IFNγ ELISpot assay, breakthrough infection

## Abstract

The effectiveness of COVID-19 vaccines developed against the original virus strain deteriorated noticeably in efficacy against the Omicron variant (B.1.1.529). Moreover, the immunity developed after vaccination or due to natural infection rapidly waned. In the present study, covering this period, we summarize the incidence of breakthrough infections among healthcare workers (HCWs) with respect to administration of the three vaccine doses. Additionally, we evaluate the long-term SARS-CoV-2-specific humoral and T cell responses at two different time points: six and twelve months after receipt of the third (booster) dose. The spike-protein-specific antibody levels and the quantity of structural-protein-specific T cells were evaluated at these time points and compared with the values measured earlier, 14 days after the booster vaccination. The study participants were categorized into two cohorts: Members of the first cohort received a two-dose BNT162b2 mRNA-based vaccine regimen, followed by an additional BNT162b2 booster six months later. Individuals in the second cohort received an inactivated-virus-based BBIBP-CorV booster six months after the initial two-dose BNT162b2 vaccination. Overall, 64.3% of participants were infected with SARS-CoV-2 confirmed by PCR or antigen test; however, additional subjects from the first cohort (23%) who did not know about their previous infection but had an anti-nucleocapsid T cell response were also considered virus-experienced. According to our results, no statistically significant difference was found between the two cohorts regarding the SARS-CoV-2-specific T cell response, neutralizing anti-RBD IgG, and anti-S IgA serum antibody levels either six or twelve months after receiving the booster, despite the overall higher median values of the first cohort. The only significant difference was the higher anti-S1/S2 IgG antibody level in the first cohort one year after the BNT162b2 booster (*p* = 0.039). In summary, the BNT162b2 and BBIBP-CorV boosters maintain durable humoral and T cell-mediated immune memory even one year after application. Although the booster provided limited protection against Omicron breakthrough infections, as 73.6% of these infections occurred after the booster vaccination, which means 53.5% cumulative incidence, it still offered excellent protection against severe disease and hospitalization in both cohorts.

## 1. Introduction

The severe acute respiratory coronavirus 2 (SARS-CoV-2) appeared in December 2019 in Wuhan, China, and as a result of its rapid spread, caused the 2019 worldwide coronavirus epidemic (COVID-19). With the further appearance of new variants of concern (VOCs), such as Alpha, Beta, and Gamma, the Hungarian government proposed from August 2021 to administer a third vaccine dose primarily to the most-risked population, including healthcare workers, to further reduce the spread of the epidemic. At this time, at least six months had passed since the two-dose immunization, and more and more evidence indicated that the immune response induced by two doses of the BNT162b2 vaccine (Pfizer-BioNTech, Mainz, Germany) would no longer be sufficient to avoid another wave of epidemics [1,2]. Therefore, HCWs participating in this study received a booster dose of the BNT162b2 or the BBIBP-CorV (Sinopharm, Beijing, China) vaccine in September 2022. Even so, a significant part of them were infected directly in the following period, when the Delta and Omicron variants caused an epidemic wave that resulted in many more confirmed COVID-19 cases than before.

Since then, several subvariants of Omicron emerged and spread worldwide. By September 2023, nearly 771 million COVID-19 cases and more than 6.9 million deaths had been reported worldwide, although the exact numbers are possibly much higher [3]. The pandemic resulted in considerable efforts worldwide to develop and introduce efficient vaccines, resulting in more than 13.5 billion doses administered worldwide as of September 2023 [4]. Nonetheless, the presently circulating Omicron XBB descendant lineages such as XBB 1.5, XBB 1.16, XBB 1.9, XBB 2.3, and EG.5.1, and, additionally, the CH.1.1 variant, a descendant of the Omicron BA.2.75, are rapidly spreading in many European countries [5]. These subvariants carry additional spike mutations that provide a substantially higher angiotensin-converting enzyme 2 (ACE2) binding affinity and impaired binding ability of antibodies induced by the first-generation booster vaccines, making these descendant lineages extremely transmissible and remarkably evasive against neutralization [6,7,8,9,10]. Another concern is that the immunity acquired by natural SARS-CoV-2 infection or vaccination wanes over time. Although previous studies have evaluated the durability of the immune response after administering the booster dose, the observation period was generally six months or less; moreover, these studies were mainly limited to monitoring only the antibody response [11,12,13,14,15]. Additionally, only a few studies have reported the durability of the spike-protein-specific T cell immunity so far, with a 6-month follow-up or less after the booster dose [16,17,18,19]. Longitudinal studies using more extended monitoring periods, eight months and twelve months, respectively, were also limited to anti-S IgG [20] or anti-RBD IgG [21] antibody levels. Therefore, more extended follow-up periods are needed to assess antibody levels in individuals after the booster dose, and, at the same time, determining the long-term virus-specific T cell memory and the incidence of breakthrough infections would also provide valuable knowledge.

Previously, we demonstrated that a BNT162b2 or BBIBP-CorV booster six months following the initial two-dose BNT162b2 or BBIBP-CorV vaccination significantly increased the anti-S1/S2 IgG, anti-S IgA, and neutralizing anti-RBD IgG serum antibody levels and induced a robust SARS-CoV-2-specific T cell response [22]. As a continuation of this work, we determine, in this paper, the humoral and T cell-mediated immune memory six and twelve months after the administration of the BNT162b2 or BBIBP-CorV booster in HCWs previously immunized with a two-dose BNT162b2 vaccine regimen. Furthermore, we evaluate the frequency and severity of breakthrough infections of Omicron variants after obtaining the booster based on positive PCR and/or antigen tests in the case of all participants and, additionally, the frequency of asymptomatic infections according to the positive anti-nucleocapsid (anti-N) T cell response in those participants immunized with three doses of the BNT162b2 vaccine.

## 2. Materials and Methods

### 2.1. Ethical Aspects of This Study

This study was conducted according to the guidelines of the Declaration of Helsinki and approved by the Ethics and Scientific Committee of the Central Hospital of Southern Pest—National Institute of Hematology and Infectious Diseases. The study participants were selected based on voluntary application, and informed consent was obtained from all subjects.

### 2.2. Study Population

All participants had received the first and second doses of the mRNA-based BNT162b2 vaccine 21 days apart between January and March 2021. Six to seven months later, between 1 and 30 September 2021, they received a booster shot, on the basis of which two cohorts were formed. Members of the first cohort received an mRNA-based BNT162b2 booster, while members of the second cohort received an inactivated-virus-based BBIBP-CorV booster (Table 1).

### 2.3. Assessment of SARS-CoV-2 Infections

All participants in this study were screened for SARS-CoV-2 infection through PCR tests between 1 September 2020 and 31 September 2022. A PCR assay was also performed in case of symptoms suspicious for COVID-19. Breakthrough infections were confirmed via Allplex SARS-CoV-2 PCR Assay (Seoul, Republic of Korea) whenever feasible or via the CLINITEST Rapid COVID-19 Antigen Test (SIEMENS Healthineers, Erlangen, Germany). A confirmed SARS-CoV-2 breakthrough infection was registered if the PCR assay or the antigen test was positive. Individuals were considered virus-naive in the first cohort if their regular PCR screening was negative (i), their SARS-CoV-2 PCR assay was negative in cases of symptoms suspicious for COVID-19 (ii), and their anti-nucleocapsid T cell response was also negative in all sample analyses (iii). The same conditions were applied to the participants of the second cohort, except for the negative anti-N T cell response, since the BBIBP-CorV booster induces a robust anti-N T cell response similar to natural infection. The inclusion criteria were the prior administration of the initial two-dose BNT162b2 vaccine regimen, followed by the BNT162b2 or BBIBP-CorV booster six months later, and voluntary registration. The exclusion criteria involved known respiratory disease (COPD, asthma), autoimmune disease, and cardiovascular disease.

With the Allplex multiplex real-time PCR assay, four target genes of SARS-CoV-2 were detected: RdRP, S (spike), N (nucleocapsid), and E (envelope). Viral RNA of SARS-CoV-2 was detected in nasopharyngeal swab specimens. The positive controls and exogenous internal controls are provided for the PCR assay kit. Real-time PCR assays were performed using a Bio-Rad modular thermal cycler platform, including a C1000 thermal cycler chassis and a CFX96 optical reaction module applying the following thermal profile: 50 °C for 20 min, 95 °C for 15 min, followed by 94 °C 15 s and 58 °C 30 s cycles repeated 44 times. For the CLINITEST Rapid COVID-19 Antigen Test, nasopharyngeal swab specimens were used to detect the nucleocapsid protein of SARS-CoV-2.

### 2.4. Determination of SARS-CoV-2-Specific T Cell Immunity

The SARS-CoV-2-specific T cell response was determined using the T-SPOT Discovery SARS-CoV-2 kit (Oxford Immunotec Ltd., Abingdon, UK). Venous blood samples were collected in sodium citrate vacutainer tubes (BD Biosciences, San Jose, CA, USA), and peripheral blood mononuclear cells were isolated via density gradient centrifugation using the Leucosep Kit (Oxford Immunotec) according to the manufacturer’s instruction. A total of 250,000 viable PBMCs were plated into each well of the 96-well microtiter plate of the T-SPOT Discovery SARS-CoV-2 kit, and the cells were incubated for 18–20 h with five different peptide pools specific to the spike (S1 and S2), nucleocapsid (N), membrane (M), and envelope (E) proteins, applying a negative control well and a positive control well (phytohemagglutinin) as well. Secreted IFNγ was captured by specific antibodies on the surface of the membrane, which formed the base of the well. After 18–20 h, the cells and other materials were removed by washing with PBS buffer. A second antibody, conjugated to alkaline phosphatase (ALP) and targeted to a different epitope on IFNγ, was added to each well, which bound to IFNγ captured on the membrane surface. The unbound antibody-ALP conjugates were removed by washing with PBS buffer. Finally, a soluble substrate was added to each well that was cleaved by the bound enzyme to form a spot of insoluble precipitate at the site of the activated T cells. The reaction was stopped after 7 min by washing the wells with distilled water. The IFNγ-producing activated effector T cells were quantified for all four viral antigens using the AID vSPOT ELISPOT reader (AID Autoimmun Diagnostika, Strassberg, Germany) by counting the spots in each well. The cumulative spot-forming units (SFUs) per 250,000 PBMCs were calculated as the total number of T-spots for S1, S2, N, M, and E antigens minus the background for each antigen (Appendix A).

### 2.5. Determination of SARS-CoV-2-Specific Antibody Levels

The anti-SARS-CoV-2 antibody levels were determined using commercially available test systems. Venous blood samples were collected in serum vacutainer tubes (BD Biosciences) and centrifuged for 10 min at 3500× *g* at room temperature. The serum samples were stored at −80 °C until measurements, if necessary. The IgG antibody levels against the SARS-CoV-2 receptor-binding domain (RBD) were evaluated using the SARS-CoV-2 Surrogate Virus Neutralization Test (sVNT) Kit (GenScript Biotech B. V., Leiden, The Netherlands), which measures neutralizing antibody levels in serum samples based on the competitive ELISA methodology. In the test kit, the ACE2 receptor protein is plated, and HRP (horseradish peroxidase)-labeled RBD was used for detection. HRP-RBD was incubated with serum samples with appropriate dilution for 30 min. The mixture was added to the wells pre-coated with ACE2 protein in the 96-well plates for 15 min. The wells were washed with wash solution for four cycles, and the HRP substrate was added to each well and incubated in the dark for 15 min. Stop solution was added to each well, and absorbance was read at 450 nm on the Liaison XL chemiluminescence analyzer (Diasorin, Saluggia, Italy). The average optical density (OD) of the negative control was used to calculate the inhibition %. The HRP-RBD blocked by the anti-RBD IgG antibodies in the serum sample was removed during the wash steps, decreasing the signal detected in the well. The serum samples with more neutralizing antibodies showed a lower signal intensity. The anti-S1/S2 IgG antibody levels were determined using the LIAISON SARS-CoV-2 S1/S2 IgG indirect chemiluminescence immunoassay (CLIA) test kit (Diasorin), while the serum IgA levels against SARS-CoV-2 spike protein were assessed via the SARS-2 COVID S IgA ELISA assay (EURO-IMMUN Medizinische Labordiagnostika, Lübeck, Germany) according to the manufacturer’s instructions. Emitted light and absorbance were measured using the Liaison XL Analyzer (Diasorin), respectively.

### 2.6. Statistical Analysis

Statistical analyses were performed by applying Student’s *t*-test, and *p*-values of <0.05 were considered statistically significant (* *p* < 0.05; ** *p* < 0.01). Statistical analysis was performed with GraphPad Prism 6 software (version 6.07).

## 3. Results

### 3.1. Vaccination Schedule, Sample Collection, and Infection Rate of Study Participants

The study participants had received the first and second doses of the mRNA-based BNT162b2 vaccine 21 days apart between January and March 2021. Afterward, 6–7 months later, between 1 and 30 September 2021, they received a BNT162b2 or BBIBP-CorV booster (Figure 1).

Depending on the type of booster, two cohorts were formed: Members of the first cohort were immunized with a homologous vaccination strategy and obtained a three-dose BNT162b2 vaccine regimen. In contrast, members of the second cohort were primed with two doses of BNT162b2 and then received a heterologous BBIBP-CorV booster. In contrast to our previous work regarding the humoral and T cell responses measured 14 days after the third dose, participants in this study were not classified into virus-naive (uninfected) and infected subgroups for two reasons. For one, until one year following the booster shot, around 85% of the study participants were infected, as detailed below. Secondly, not all infected individuals had an anti-N T cell response a few months later, making it impossible to identify all asymptomatic infections and recognize who was still assuredly virus-naive at the last sampling date.

Until September 2022, 1 year after receiving the third dose, 64.3% of the participants had SARS-CoV-2 infection confirmed by a PCR or antigen test. In detail, 61.5% of the participants in the first cohort had symptomatic SARS-CoV-2 infection confirmed by PCR and/or antigen tests, but, importantly, an additional 23% of subjects in this cohort had an anti-N T cell response. Since the BNT162b2 vaccine contains only the spike protein mRNA, the positive anti-N T cell response could only be elicited by contact with the SARS-CoV-2 virus. These participants did not know about previous infection and did not experience any symptoms, suggesting a history of unrecognized, asymptomatic SARS-CoV-2 infection. Altogether, 84.5% of individuals in the first cohort experienced immunogenic impacts from the beginning of the pandemic up to 12 months after obtaining the booster dose. In the second cohort, 66.6% of participants had symptomatic SARS-CoV-2 infection confirmed by a PCR and/or antigen test until one year after administering the booster dose, which is very similar to the infection rate of the first cohort. However, it was not feasible to identify asymptomatic SARS-CoV-2 infections in this cohort because the inactivated virus-based BBIBP-CorV vaccine induces a robust anti-N T cell response similar to natural infection. The rate of infected but asymptomatic participants may be comparable to that in the first cohort (~20–25%). It is necessary to emphasize that 73.6% of the breakthrough infections confirmed by a PCR or antigen test were registered in early 2022, 4–5 months after administering the booster dose, when the pandemic wave was dominated by the Omicron and partly by the Delta variant in Hungary. In summary, although the effectiveness of the BNT162b2 or BBIBP-CorV boosters against SARS-CoV-2 infection rapidly waned, importantly, all infected participants experienced mild symptoms, and none of them had severe COVID-19 infection leading to hospitalization.

### 3.2. SARS-CoV-2-Specific T Cell Response Six and Twelve Months after Administering BNT162b2 or BBIBP-CorV Boosters

The SARS-CoV-2-specific T cell response was determined six and twelve months after the booster and compared with those levels measured earlier, 14 days after obtaining the BNT162b2 or BBIBP-CorV booster. The cumulative IFNγ-positive T cell response was determined considering the reactivity with all structural SARS-CoV-2 proteins, including the spike (S1 and S2), nucleocapsid (N), membrane (M), and envelope (E).

Despite the considerable differences between the two cohorts being measured shortly after the third dose administration (medians of 166 SFU/2.5 × 10^5^ PBMC in the first and 84 SFU/2.5 × 10^5^ PBMC in the second cohort; *p* = 0.024), no statistically significant difference was found between the two cohorts regarding the cumulative T cell response either six (medians of 82 SFU/2.5 × 10^5^ PBMC in the first and 101 SFU/2.5 × 10^5^ PBMC in the second cohort; *p* = 0.380) or twelve (medians of 93.5 SFU/2.5 × 10^5^ PBMC in the first and 74 SFU/2.5 × 10^5^ PBMC in the second cohort; *p* = 0.930) months later (Figure 2).

Within the first (BNT162b2 booster) cohort, six months after administering the third dose, the cumulative T cell response significantly decreased from a median value of 166 SFU/2.5 × 10^5^ PBMC to 82 SFU/2.5 × 10^5^ PBMC (*p* = 0.012), then increased slightly after one year to a median value of 93.5 SFU/2.5 × 10^5^ PBMC (*p* = 0.465). At this time point, the cumulative T cell response was not significantly lower than 14 days after the booster vaccination (*p* = 0.057). Within the second (BBIBP-CorV booster) cohort, six months after receiving the third dose, the cumulative T cell response slightly increased from a median value of 84 SFU/2.5 × 10^5^ PBMC to 101 SFU/2.5 × 10^5^ PBMC, then decreased after one year to a median value of 74 SFU/2.5 × 10^5^ PBMC. However, the changes in the cumulative T cell responses among the three monitoring time points were not statistically significant.

### 3.3. SARS-CoV-2-Specific Antibody Response Six and Twelve Months after Administering BNT162b2 or BBIBP-CorV Boosters

The SARS-CoV-2-specific antibody responses, including the anti-S1/S2 IgG, neutralizing anti-RBD IgG, and anti-S IgA serum antibody levels, were determined six and twelve months after administering the booster dose and compared with the antibody levels measured earlier, fourteen days after the booster vaccination. Regarding the anti-S1/S2 IgG antibody levels, we found that the significant difference between the two cohorts 14 days after the booster injection (medians of 6000 AU/mL in the first and 282 AU/mL in the second cohort; *p* = 0.0024) was diminished after six months, and the median antibody levels of the two cohorts became nearly equal (medians of 2270 AU/mL in the first and 3490 AU/mL in the second cohort; *p* = 0.178) (Figure 3).

However, 12 months after the booster vaccination, the median anti-S1/S2 IgG antibody level of the first cohort was significantly higher again than that of the second cohort (medians of 5515 AU/mL in the first and 2070 AU/mL in the second cohort; *p* = 0.039). Within the first cohort, six months after the booster dose, the anti-S1/S2 IgG antibody level decreased from a median value of 6000 AU/mL to 2270 AU/mL, then increased one year following the booster to a median value of 5515 AU/mL, but the alterations among the three monitoring time points were not statistically significant. Within the second cohort, six months after the booster dose, the anti-S1/S2 IgG antibody level significantly increased from a median value of 282 AU/mL to 3490 AU/mL (*p* = 0.047), then decreased slightly after one year to a median value of 2070 AU/mL (*p* = 0.243). These levels measured one year after the booster were markedly but not statistically significantly higher than 14 days after the booster dose administration (*p* = 0.067).

Although a robust difference was measured between the two cohorts regarding the neutralizing anti-RBD IgG serum antibody level 14 days after the booster vaccination (medians of 98.5% in the first and 85.8% in the second cohort; *p* = 0.019), this difference was no longer statistically significant six (medians of 98.8% in the first and 98.7% in the second cohort; *p* = 0.074) and twelve months later (medians of 98.5% in the first and 98.4% in the second cohort; *p* = 0.450) (Figure 4).

Within the first cohort, the median level of the neutralizing antibodies remained highly stable at the three monitoring time points (98.5%, 98.8, and 98.4%, respectively). In contrast, an increasing trend was observed within the second cohort (85.8%, 98.7%, and 98.4%, respectively). However, these changes among the three monitoring time points were not statistically significant in either cohort.

Finally, the spike-protein-specific IgA serum antibody levels were also significantly higher in participants in the first cohort 14 days after the booster vaccination (medians of 13.5 S/CO in the first and 5 S/CO in the second cohort; *p* = 0.001) (Figure 5).

However, the difference between the two cohorts six (medians of 10.6 S/CO in the first and 9.5 S/CO in the second cohort; *p* = 0.811) and twelve months later (medians of 9.2 S/CO in the first and 9.7 S/CO in the second cohort; *p* = 0.936) was not statistically significant. At these time points, an almost identical anti-S IgA response was detected in the first and second cohorts. During follow-up, a decreasing trend was observed in the first cohort: the median anti-S IgA antibody level significantly decreased from 13.5 S/CO to 10.6 S/CO after six months (*p* = 0.021), then slightly decreased to a median value of 9.2 S/CO after twelve months (*p* = 0.960). One year after the booster vaccination, the anti-S IgA antibody level was significantly lower than 14 days after receiving the third dose (*p* = 0.007). In contrast, a slightly increasing trend was observed in the second cohort (5 S/CO, 9.5 S/CO, 9.7 S/CO), but the alterations among the three monitoring time points were not statistically significant.

## 4. Discussion

Six to seven months after administering the initial two-dose BNT162b2 vaccination, the study participants received either an mRNA-based BNT162b2 (first cohort) or an inactivated virus-based BBIBP-CorV (second cohort) booster between 1 and 30 September 2021. During their follow-up, we evaluated the spike protein-specific antibody levels, including the anti-S1/S2 IgG, neutralizing anti-RBD IgG, and anti-S IgA serum antibody levels, and, additionally, the spike-, nucleocapsid-, membrane-, and envelope-specific T cell responses at two different time points, six and twelve months after receiving the booster. These antibody levels and cumulative T cell numbers were compared with the values measured earlier, 14 days after the booster vaccination.

All participants in this study were subjected to regular monitoring for SARS-CoV-2 infection between 1 September 2020 and 30 September 2022. Until one year after the booster dose administration, 61.5% of the participants in the first cohort had symptomatic SARS-CoV-2 infection confirmed by a PCR or antigen test, and an additional 23% of individuals had an anti-N T cell response. As the BNT162b2 vaccine encodes only the spike protein, the positive anti-N T cell response suggests a history of unrecognized, asymptomatic SARS-CoV-2 infection. A similar infection rate was observed in the second cohort, as 66.6% of participants had symptomatic SARS-CoV-2 infection verified by PCR or antigen test from the beginning of the pandemic up to one year after obtaining the booster dose. However, as the inactivated-virus-based BBIBP-CorV booster induces an anti-N T cell response similar to natural SARS-CoV-2 infection, it was inconceivable to identify individuals with previous asymptomatic SARS-CoV-2 infections in this cohort. The proportion of infected but asymptomatic participants was estimated to be similar to that of the first group (~20–25%). Importantly, 73.6% of the confirmed breakthrough infections were registered in January and February 2022, 4–5 months after obtaining the booster dose. Based on these data, the cumulative incidence was 53.5% after the booster vaccination among our study participants. In this period, the dominant SARS-CoV-2 variants were the Omicron and the Delta variants in Hungary, culminating in a staggering number of confirmed COVID-19 cases, hospitalizations, and deaths. The most likely reason for the high incidence of breakthrough infections was the inadequate stringency of non-pharmacological prevention and control measures, including testing, contact tracing, home isolation, mask wearing, proper social distancing, and travel restrictions. These results are highly similar to those of other groups, where the cumulating incidence of breakthrough infection was 41.8% and 47% among HCWs after the third BNT162b2 vaccination, and the follow-up periods were eight months and four months, respectively, which are two-thirds and one-third of ours [23,24]. Although a more extensive European multicentric longitudinal cohort study involving 63,516 HCWs within the ORCHESTRA project estimated the cumulative incidence of symptomatic breakthrough infections to be even lower, with 20.3–30% in HCWs receiving the booster dose, importantly, the median follow-up period was only six months, half of ours, making these results entirely comparable with ours [25]. It is essential to emphasize that none of our study participants had severe COVID-19 infections, and this is consistent with the results of large cohort studies in Hungary investigating the effectiveness of different vaccination series against severe SARS-CoV-2 infection and hospitalization [26,27].

Our results show that the magnitude of cumulative T cell response was twice as high in the first than in the second cohort 14 days after receiving the booster dose. However, no statistically significant difference was found between the two cohorts during the long-term follow-up, nor six or twelve months after the booster dose administration. Analyzing the alteration in the T cell response within the cohorts over time, a significant change occurred only in the first cohort, as six months after the booster vaccination, the number of T cells reacting with the viral peptides decreased significantly. However, one year following the booster vaccination, the cumulative number of virus-specific T cells increased to a similar level to that measured two weeks after receiving the booster, and thus the T cell response remained quite stable up to one year following the third dose. These results are in line with the findings of other groups showing that in BNT162b2-primed individuals, the T cell-mediated response was similarly robust and well maintained for several months after either a BNT162b2 [16] or mRNA-1273 (Moderna, Cambridge, MA, USA) booster [19]. In a comparison of the homologous vaccination composed of three doses of the BNT162b2 vaccine and the heterologous vaccination utilizing two doses of AZD1222 (Oxford/AstraZeneca, Cambridge, UK) followed by a BNT162b2 booster, comparable results were also reported, as the T cell response remained vigorous and similar in both cohorts up to six months after the third vaccination [17].

Concerning the anti-S1/S2 IgG antibody levels, significant differences were also found between the two cohorts 14 days after the booster injection (*p* = 0.0024). The participants in the first cohort had more than 20 times higher antibody levels than the participants in the second cohort who received a BBIBP-CorV booster. Six months later, the anti-S1/S2 IgG antibody levels of the two cohorts became very similar because of the robust elevation of the antibody levels in the second cohort (from a median of 282 AU/mL to 3490 AU/mL), while one year after the booster shot, significantly higher antibody levels were measured again in the first cohort. The neutralizing anti-RBD IgG and serum anti-S IgA antibody levels were significantly higher in the first cohort 14 days after administering the third dose, but there was no significant difference between the two cohorts after either six or twelve months. Regarding the temporal change in the humoral immune response, significant changes were found in both cohorts. The participants who received a heterologous BBIBP-CorV booster had remarkably higher anti-S1/S2 IgG antibody levels six months later compared to fourteen days after receiving the third dose (*p* = 0.047). Although the neutralizing anti-RBD IgG levels remained stable during the follow-up in both cohorts, the anti-S IgA serum levels were much lower in the first cohort both after six (*p* = 0.021) and twelve months (*p* = 0.007) compared to the first time point, fourteen days after the BNT162b2 booster vaccination. These results are consistent with the tendency found by other groups, according to which the remarkable anti-S IgG and anti-RBD IgG antibody levels persist up to twelve months after booster vaccination. The healthcare workers participating in these studies received three doses of the BNT162b2 or mRNA-1273 vaccines or received heterologous vaccination, two doses of the BNT162b2 vaccine followed by an mRNA-1273 booster, or two doses of the CoronaVac (Sinovac, Beijing, China) vaccine followed by an mRNA-1273 booster [20,21]. Nevertheless, the high rate of breakthrough infections (84.5% in the first cohort) could have caused the sustained elevated anti-S1/S2 IgG and anti-RBD IgG antibody levels we measured, as hybrid immunity is known to provide increased and long-lasting antibody levels compared to virus-naive vaccinees [28,29].

The main limitations of our work include a small sample size, as only 28 healthcare workers could be recalled for the follow-up, with 13 individuals in the first cohort and 15 in the second cohort. Another limitation of the study is identifying only the SARS-CoV-2-responsive IFN-γ secreting T cells, and not investigating T cell subsets or other non-IFN-γ secreting immune cells.

## 5. Conclusions

In conclusion, vaccination with BNT162b2 or BBIBP-CorV boosters induced a robust and durable humoral and T cell-mediated immune memory enduring for at least one year. However, the protection of the third dose against breakthrough infections caused by the Delta or Omicron variants was insufficient 4–5 months after administration, and the high rate of SARS-CoV-2 breakthrough infections may have played a significant role in sustained elevated antibody levels and T cell responsiveness.

## Figures and Tables

**Figure 1 vaccines-12-00003-f001:**
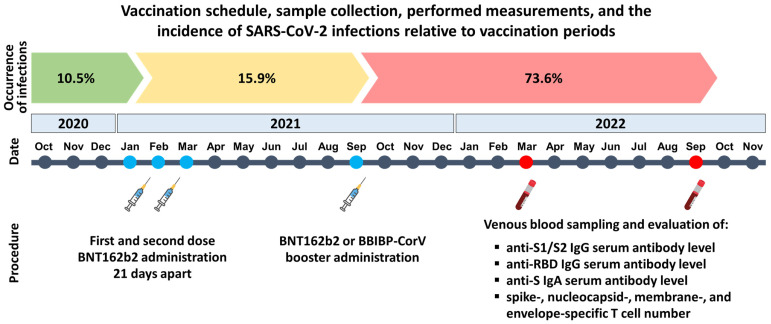
Vaccination schedule, sample collection, and measurements performed in this study. Participants received a BNT162b2 or BBIBP-CorV booster 6–7 months after the initial two-dose BNT162b2 vaccine regimen. Samples were collected and analyzed 6 and 12 months after the booster dose administration. Symptomatic infections were confirmed by positive PCR and/or antigen tests. A total of 10.5% of study participants became infected before the first dose, 15.9% between receiving the second and third dose, and 73.6% after obtaining the booster dose. The cumulative IFNγ-positive T cell number (total SFU against S1, S2, N, M, and E viral proteins) was determined via the T-SPOT Discovery SARS-CoV-2 ELISpot assay. The spike-specific anti-S1/S2 IgG, anti-RBD IgG, and anti-S IgA serum antibody levels were evaluated using the LIAISON SARS-CoV-2 S1/S2 IgG test, the SARS-CoV-2 surrogate virus neutralization test (sVNT), and the SARS-CoV-2 anti-S IgA assay, respectively.

**Figure 2 vaccines-12-00003-f002:**
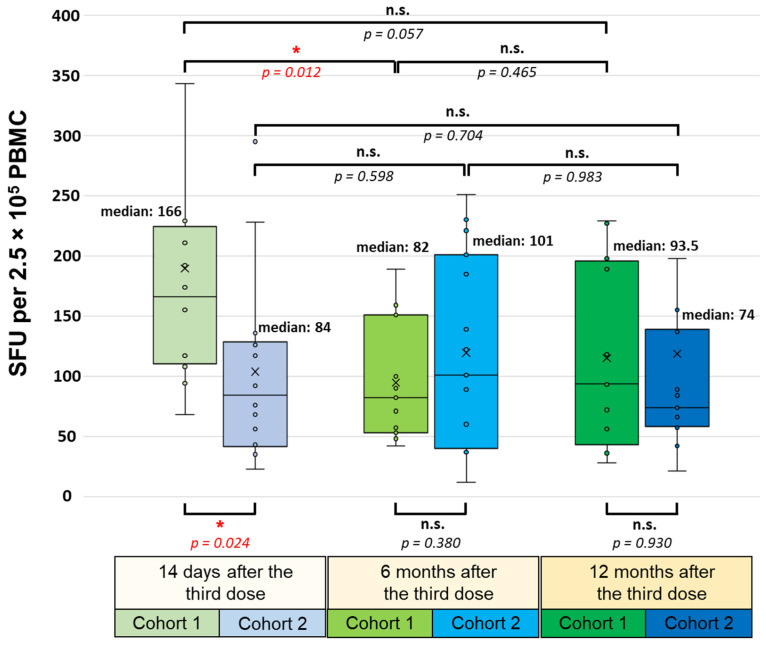
SARS-CoV-2-specific T cell response 14 days, 6 months, and 12 months after administering the third (booster) dose of the BNT162b2 (mRNA-based) or the BBIBP-CorV (inactivated-virus-based) vaccine. All participants were immunized with an initial two-dose BNT162b2 vaccine regimen 6–7 months before receiving the booster. The cumulative IFNγ-positive T cell responses were evaluated via the T-SPOT Discovery SARS-CoV-2 ELISpot assay and calculated as the total SFU against S1, S2, N, M, and E antigens. Statistical analyses were performed by applying Student’s *t*-test and *p*-values of <0.05 were considered statistically significant (*), while *p*-values of >0.05 were considered non-significant (n.s.). Box plots display the median values with the interquartile range (lower and upper hinges) and ±1.5-fold of the interquartile range from the first and third quartiles (lower and upper whiskers).

**Figure 3 vaccines-12-00003-f003:**
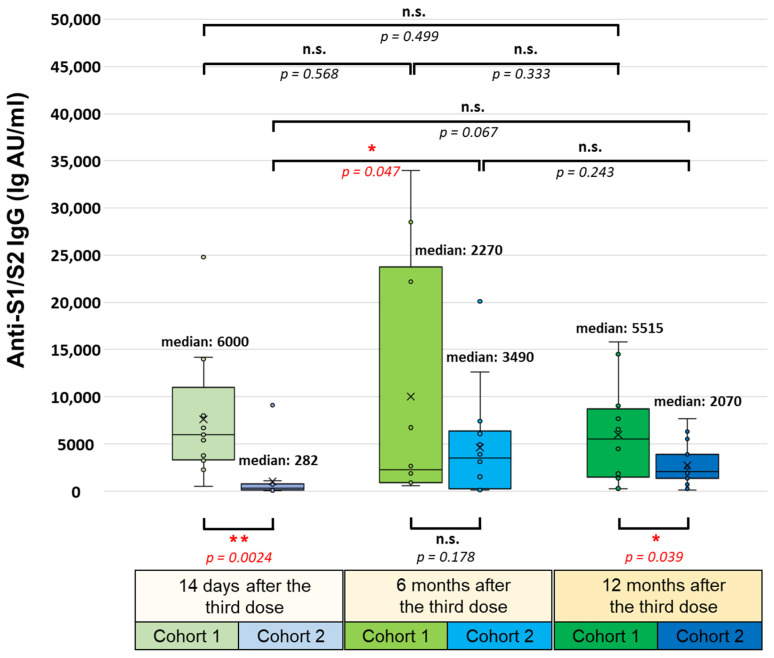
SARS-CoV-2-specific anti-S1/S2 IgG response 14 days, 6 months, and 12 months after administering the third (booster) dose of the BNT162b2 (mRNA-based) or the BBIBP-CorV (inactivated-virus-based) vaccine. All participants were immunized with an initial two-dose BNT162b2 vaccine regimen 6–7 months before receiving the booster. The SARS-CoV-2-specific anti-S1/S2 IgG levels were determined via the LIAISON SARS-CoV-2 S1/S2 IgG test (Diasorin S.P.A.). Statistical analyses were performed by applying Student’s *t*-test and *p*-values of <0.05 were considered statistically significant (* *p* < 0.05; ** *p* < 0.01), while *p*-values of >0.05 were considered non-significant (n.s.). Box plots display the median values with the interquartile range (lower and upper hinges) and ±1.5-fold of the interquartile range from the first and third quartiles (lower and upper whiskers).

**Figure 4 vaccines-12-00003-f004:**
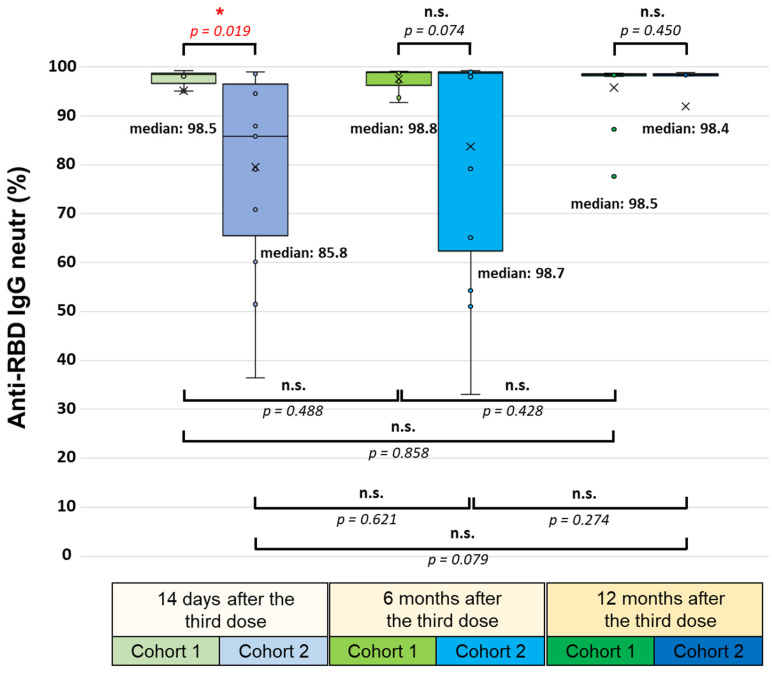
SARS-CoV-2-specific neutralizing anti-RBD IgG levels 14 days, 6 months, and 12 months after administering the third (booster) dose of the BNT162b2 (mRNA-based) or the BBIBP-CorV (inactivated-virus-based) vaccine. All participants were immunized with an initial two-dose BNT162b2 vaccine regimen 6–7 months before receiving the booster. The SARS-CoV-2-specific anti-RBD IgG levels were determined via the SARS-CoV-2 surrogate virus neutralization test (sVNT) (GenScript Biotech B.V.). Statistical analyses were performed by applying Student’s *t*-test, and *p*-values of <0.05 were considered statistically significant (*), while *p*-values of >0.05 were considered non-significant (n.s.). Box plots display the median values with the interquartile range (lower and upper hinges) and ±1.5-fold of the interquartile range from the first and third quartiles (lower and upper whiskers).

**Figure 5 vaccines-12-00003-f005:**
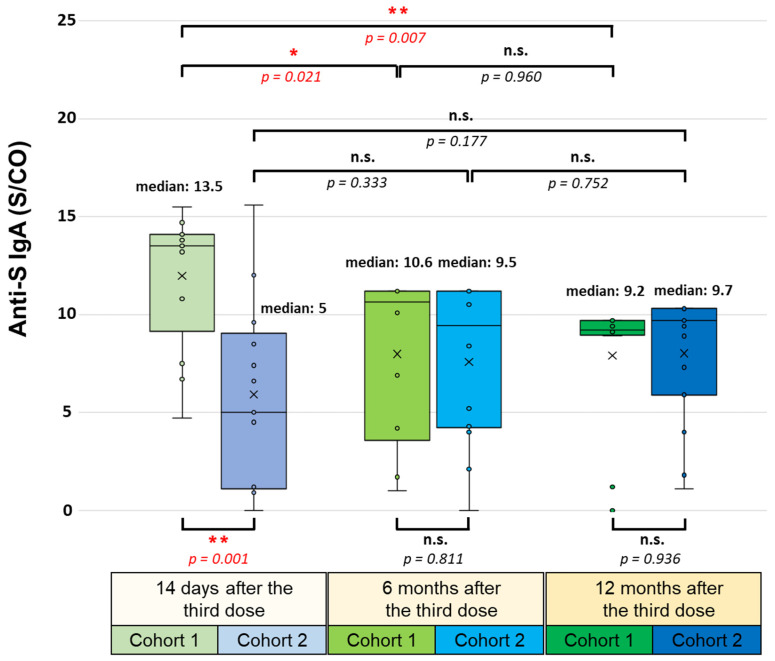
SARS-CoV-2-specific anti-S IgA serum levels 14 days, 6 months, and 12 months after administering the third (booster) dose of the BNT162b2 (mRNA-based) or the BBIBP-CorV (inactivated-virus-based) vaccine. All participants were immunized with an initial two-dose BNT162b2 vaccine regimen 6–7 months before receiving the booster. The SARS-CoV-2-specific anti-S IgA serum levels were determined via the SARS-CoV-2 IgA assay (EURO-IMMUN Medizinische Labordiagnostika AG). Statistical analyses were performed by applying Student’s *t*-test, and *p*-values of <0.05 were considered statistically significant (* *p* < 0.05; ** *p* < 0.01), while *p*-values of >0.05 were considered non-significant (n.s.). Box plots display the median values with the interquartile range (lower and upper hinges) and ±1.5-fold of the interquartile range from the first and third quartiles (lower and upper whiskers).

**Table 1 vaccines-12-00003-t001:** Characteristics of study participants.

	Cohort 1	Cohort 2
Study participants	Laboratory technicians (77%) and physicians (23%)	Laboratory technicians (54%) and physicians (46%)
Number of individuals	13	15
Vaccine combinations	BNT162b2 + BNT162b2 + BNT162b2	BNT162b2 + BNT162b2 + BBIBP-CorV
Age (median; age range)	48 (38–61)	47 (35–64)
Gender		
Female	12	13
Male	1	2

## Data Availability

All data, materials, and methods used in the analysis are available from the corresponding author on request.

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
