# Peer review of "Long-Term SARS-CoV-2-Specific Humoral and T Cell Responses after the BNT162b2 or BBIBP-CorV Booster and the Incidence of Breakthrough Infections among Healthcare Workers"

_vaccines, 2023, doi:10.3390/vaccines12010003_

Round 1

Reviewer 1 Report

Comments and Suggestions for Authors

The manuscript by study entitled “Long-Term SARS-CoV-2-Specific Humoral and T cell Responses after the BNT162b2 or BBIBP-CorV Booster and the Incidence of Breakthrough Infections among Healthcare Workers” evaluated the SARS-CoV-2-specific humoral and T cell responses of healthcare workers at six and twelve months after receiving the third (booster) dose. The incidence of breakthrough infections among the cohorts was also estimated. The data brings important information to the literature since different vaccination schemes have shown to induce distinct immune response profiles. 

Comments on the Quality of English Language

The manuscript is written in an informative, concise and clear format. Minor English revision is recommended.

Author Response

Dear Reviewer,

Please see the attachment, which contains the review report alongside our comprehensive responses addressing each point.

Sincerely yours,
The Authors

Reviewer 2 Report

Comments and Suggestions for Authors

The reviewer has revision suggestions as follow. 

The tests for the detection of SARS-CoV-2 infection provide critical results for protectiveness analysis. Could the authors detail the method of these tests in Method section?

Two cohorts has been described from line 140 to 143. It is strongly suggested to show cohort 1 and 2 instead of the names of commercial vaccines in the figures, and also instead of BNT162b2 and BBIBP cohorts in the article. 

In line 145 to 146, authors stated that participants had also been subgrouped into virus-naive uninfected and infected, but the data on their immune responses was absent in the article. 

In line 367, the name SAR-CoV-2 is incomplete. 

The content "... the booster vaccination still provided excellent protection ... among our study participants" in line 370 and 371 should be moved to Discussion section, if appropriate, but it should not be a conclusion, as no related results were found in this article. 

Comments on the Quality of English Language

no comments.

Author Response

(The authors gave the same response as above.)

Reviewer 3 Report

Comments and Suggestions for Authors

- In line 105 and 106, The IFNγ-producing activated T cells were quantified for all four viral anti-gens using the AID vSPOT ELISPOT reader.

: Could authors show about “The IFNγ-producing activated T cells'"? 

- Authors described the Virus Neutralization Test. 

In line 112 and 114, The serum samples were stored at -80 °C until measurements if needed. IgG antibody levels against SARS-CoV-2 receptor-binding domain (RBD) were evaluated by the SARS-CoV-2 Surrogate Virus Neutralization Test (sVNT) Kit (GenScript Biotech B. V., Leiden, Netherlands).

In line 212 and 215, SARS-CoV-2-specific antibody response, including anti-S1/S2 IgG, neutralizing anti-RBD IgG, and anti-S IgA serum antibody levels, were determined six and twelve months after administering the booster dose and were compared with antibody levels measured earlier, 14 days after the booster vaccination.

: Could authors show more detail about neutralizing results? 

- Could authors also show data of anti-N IgG and anti-N IgA serum antibody levels? Because authors also comment about “the Incidence of Breakthrough Infections among Healthcare Workers” .

In line 365 and 368, In conclusion, vaccination with BNT162b2 or BBIBP-CorV boosters induced a robust and durable humoral and T cell-mediated immune memory enduring for at least one year; however, the high rate of SAR-CoV-2 infections certainly played a significant role in sustained elevated antibody levels and T cell responsiveness.

: Could authors suggest what is a good vaccine between BNT162b2 and BBIBP-CorV boosters about induction of a robust and durable humoral and T cell-mediated immune memory?

- Could authors put the full name of HCWs in line 51? 

- Could authors put the full name of ACE2 in line 69? 

Comments on the Quality of English Language

NOTHING TO COMMENTS. 

Author Response

(The authors gave the same response as above.)
